# Multi-Scale Characterisation and Mechanical Adhesion in PVD-Deposited Ca-SZ Coating for Implantable Medical Devices

**DOI:** 10.3390/biomedicines13010037

**Published:** 2024-12-27

**Authors:** Alex Tchinda, Richard Kouitat-Ndjiwa, Pierre Bravetti

**Affiliations:** Jean Lamour Institute, Department of Micro and Nanomechanics for Life, University of Lorraine, UMR 7198, 54011 Nancy, Francepierre.bravetti@univ-lorraine.fr (P.B.)

**Keywords:** coating, Ca-SZ, adhesion, biomechanics, Young’s modulus

## Abstract

Oral implantology faces a multitude of technical challenges in light of current clinical experience, underlining the need for innovation in implantable medical devices in both mechanical and biological terms. **Objectives:** This study explores the influence of the thickness factor of calcium-doped zirconia (Ca-SZ) coatings deposited by PVD on their intrinsic mechanical properties and the determinism of the latter on adhesion to the TA6V alloy substrate after mechanical loading for applications in dental implantology. **Methods:** Three separate thicknesses of 250 nm, 450 nm and 850 nm were evaluated in terms of mechanical strength, modulus of elasticity and adhesion to the substrate, in accordance with ISO 20502:2005. **Results**: The results show an increase in apparent modulus of elasticity with thickness, reaching values of around 25.05 GPa and 36.3 GPa, close to the cortical bone for the 250 nm and 450 nm thick coatings, respectively. Adhesion tests show a progressive improvement up to 450 nm, followed by a similar observation at 850 nm, underlining the importance of optimal thickness to balance mechanical protection and biomechanical compatibility. Furthermore, the initial roughness and topography of the substrate were not influenced by the different thicknesses of the Ca-SZ coating. **Conclusions:** Together, these results reinforce the potential of Ca-SZ coatings to minimise stress shielding in dental implants.

## 1. Introduction

Modern oral implantology is facing a major turning point in terms of innovation. Following the many clinical obstacles observed, designers are redoubling their inventiveness to overcome the difficulties associated with the progressive increase in peri-implantitis, the prevalence of which stands at 43.9% for patients without preventive maintenance, according to Costa et al. [1]. Although prevalence rates vary from study to study and from one geographical region to another, they are nonetheless very worrying. The common denominator observed in these observations is frequently the use of titanium alloys as implant material, which, according to a very recent study, has been shown to be a determining factor in the risk of titanium particles being released at the periphery of the implant, leading to peri-implant osteolysis [2]. Furthermore, a simulation study of stress distribution using finite element analysis demonstrated a homogeneous stress distribution around the cortical bone for zirconia implants, thereby minimising the risk of stress shielding, which is responsible for failures due to peri-implant bone resorption, unlike titanium implants [3]. However, the literature widely supports the use of materials with a low Young’s modulus compared to titanium, such as PEEK, to improve stress distribution in the bone in order to optimise mechanical biocompatibility and rapid osseointegration [4,5,6]. However, the common thread running through the various mechanical and biological parameters conditioning successful osseointegration remains the homeostatic bone remodelling process. Indeed, in response to the perpetual mechanical stresses to which bone is subjected, physiological feedback processes operate in such a way as to continually readjust the intrinsic properties of exposed bone, such as density, geometry and three-dimensional architecture, in line with the functional adaptation thresholds according to Frost’s mechanostat [7]. The bone matrix is a dynamic system in constant search of equilibrium, frequently modulated by various environmental constraints. However, optimising the osseointegration process for new-generation implants requires the cross-involvement of various factors, such as choosing the right material with the optimum mechanical parameters to minimise the risk of failure caused by the above-mentioned physiological processes, in the same way as the functionalisation of surfaces to initiate bone healing, this time in a proactive manner [8]. In this context, this study explores the design of a highly innovative hybrid dental implant with a Ca-SZ coating deposited by PVD on a TA6V substrate. The idea is to benefit from the advantages of these materials while avoiding their respective disadvantages. More specifically, the aim is to benefit, on the one hand, from the ductility of titanium, which makes up the body of the implant, while at the same time benefiting on the surface from the excellent biocompatibility of zirconia, which is superior to titanium according to an in vivo study [9]. In addition, this design would confine the release of titanium particles, thereby eliminating peri-implant osteolysis caused by titanium [2]. As an added bonus, calcium doping of zirconia in the Ca-SZ crystallographic configuration aims to functionalise surfaces by chemotaxis activation of osteocytes responsible for bone remodelling, making the surface bioactive and proactive for an optimal osteogenic response. [10,11,12] More importantly, the presence of zirconia on the surface also acts as an effective shield to limit bacterial biofilm colonisation and proliferation, thereby eliminating the risk of peri-implantitis [13,14,15] Consequently, this study incorporates multi-scale characterisation techniques for the mechanical properties of the Ca-SZ coating, with the emphasis on analysing the adhesion of the coating/substrate pair in order to draw up a reliability profile for this innovative coating.

## 2. Materials and Method

### 2.1. Ca-SZ Reactive Sputter Deposition

The Ca-SZ coatings were deposited by reactive magnetron sputtering onto ø 2 cm, 2 mm thick TA6V alloy substrates supplied by Visy implant (Chavanod, France). Deposits were made in a vacuum chamber equipped with two 2-inch diameter cathodes, one made of zirconium at 99.99% purity and an applied electrical power of 0.45 A, and the other made of calcium at 99.99% purity and an applied electrical power of 0.10 A. The initial vacuum was of the order of magnitude of 10^−7^ mbar. The working pressure was maintained at 1 Pa, then an argon flow rate of 30 sccm was introduced and an oxygen flow rate of 10 sccm was also introduced into the EVA300 deposition chamber (Institut Jean Lamour, Nancy, France). The deposition method and microstructural characterisation were described in detail by Tchinda et al. in their previous article [16].

### 2.2. SEM Micrograph and EDS Mapping

Discs were metallised by deposition of a 15 nm thick carbon layer by the Safematic Compact Coating unit-010 metalliser (Zizers, Switzerland) under pressure for 10 min. The discs were then observed using a Quanta FEG 650 SEM (Institut Jean L’amour, Nancy, France) between 5 and 15.00 kV. EDS mapping was then performed at 20 kV for 15 min of acquisition. The EDS profiles of the tracks were carried out along a line with 1000 measurement points.

### 2.3. Surface Topography: Optical Profilometer

The standard roughness measurement was carried out using the X10 lens of the GBS smartWLI(Ilmenau, Germany) Extended in accordance with ISO 25178 [17]. A 250 µm Gaussian filter was applied using MountainsMap Imaging Topography 7.4.8051 software. Three tests were carried out on various samples of each coating thickness. The topography of a surface measured by optical profilometry is generally characterised by the parameters Sa and Sq. The average of the absolute deviations of the heights from the mean plane is known as the arithmetic mean roughness (Sa), while the root-mean-square deviation (Sq) provides a weighted assessment of topographical variations. These values enable us to accurately assess the texture of the surface under study.

### 2.4. Ultra-Nano-Indentation (UNHT)

Measurements of mechanical parameters were obtained using the ultra-nano-indenter tester module equipped with a spherical ruby tip of radius 200 µm (Ref:RB888) from CSM Instruments (actually Anton Paar), (Baden, Switzerland). The maximum load was 20 mN for all measurements. The loading and unloading rate was 90 mN/min for a 10 s holding time with an acquisition rate of 10 Hz. N = 3 trials were selected for each sample. Analysis of the first linear points based solely on the curves *a*^3^/*R* (mm) on the abscissa and *P* (N) on the ordinate enabled us to assess the elastic properties of the Ca-SZ coating. Young’s modulus (*E*) was calculated using the relationship *k* = *E*/1 − *ν*^2^, considering *ν* = 0.3, where *k* is the contact stiffness, expressing the ratio between the applied load (*P*) and the normalized elastic displacement (*a*^3^/*R*), where a is the contact radius and *R* the radius.

### 2.5. Mechanical Adhesion by Scratch Test: ISO 20502:2005

The scratch tests were carried out using the surface testing platform STeP 4 model MST3 from Anton Paar (Institut FEMTO-ST, Besançon, France). The device was fitted with a Rockwell Sphero conical diamond tip with ρ = 100 µm, Cat No. 144994 (Anton Paar, Baden, Switzerland). The tests were carried out in accordance with ISO 20502:2005 [18] with a progressive load of 0–10N and a loading speed of 200 mN/s over a distance of 2 mm. Nine tests were carried out on various samples of each coating thickness in random directions. Critical loads were determined from the friction force curves. The critical thresholds LC1, LC2 and LC3 were established on the basis of fluctuations in the friction force curve recorded during the scratch test. The appearance of the first surface cracks in the coating is represented by LC1, while LC2 indicates an increase in internal cohesive damage, and LC3 indicates deterioration of the coating, with no signs of deterioration.

## 3. Results

### 3.1. SEM Micrography and EDS Mapping

The low-magnification SEM micrograph in Figure 1 shows an overall view of a Ca-SZ coating that has good coverage and no apparent defects after deposition. The coating perfectly matches the grooves in the underlying substrate surfaces that resulted from the substrate machining process. In parallel, the elemental EDS mapping of the Zirconium, Oxygen and Calcium constituents of the coating shows not only a homogeneous and uniform distribution of the elements but also an absence of structural defects independent of the coating thicknesses (Figure 2).

### 3.2. Surface Topography

Figure 3 illustrates the 3D surface profile of the Ca-SZ coatings, showing different thicknesses comparatively similar to the bare TA6V substrate surface. Again here, the coatings perfectly match the surface grooves of the underlying substrate. In addition, Figure 4 reveals a trend towards increasing topographic parameters Sa and Sq with increasing coating thickness that remains insignificant. Using a one-way analysis of variance (ANOVA ONE WAY) to assess the differences between the groups, for a *p*-value < 0.05 the difference would have been considered significant.

### 3.3. Ultra-Nano-Indentation

Table 1 shows the UNHT tests carried out on Ca-SZ coatings of different thicknesses deposited by PVD on a TA6V substrate. Analysis of the first linear points from the curves a^3^/R (mm) on the abscissa and *P* (N) on the ordinate made it possible to assess the elastic properties of the Ca-SZ coating. It should be noted that the values of the slopes k, obtained from these data, change progressively, rising from 36.7 × 10^3^ N/mm^2^ for 250 nm to 53.2 × 10^3^ N/mm^2^ for 450 nm, and up to 64.6 × 10^3^ N/mm^2^ for 850 nm on average. It seems clear that the various increases observed reflect an increasing apparent elasticity, confirmed by the values of *E*, calculated using the relationship *k* = 4*E*/3 (1 − *ν*^2^) assuming *ν* = 0.3. As a result, the modulus *E* increases from 25.05 GPa to 43.35 GPa for 250 nm and 850 nm, respectively. These observations suggest a key role of the thickness variant in the mechanical properties, with potentially better elastic stress dissipation in thicker layers (850 nm). Conversely, for thinner coatings (250 nm), the influence of the TA6V substrate is likely to be more pronounced, which could explain a lower apparent modulus, potentially underestimated in relation to the intrinsic properties of the coating alone.

### 3.4. Results of Mechanical Adhesion by Scratch Test: ISO 20502:2005

Figure 5 SEM micrographs of the tracks illustrate the mechanical behaviour of Ca-SZ coatings of different thicknesses: 250 nm, 450 nm and 850 nm subjected to stresses of the same nature. Figure 5A shows cohesive failure over the entire track. The presence of beads at the end of the track suggests simple deformation. The associated critical loads LC1, LC2 and LC3 are approximately 4 N, 6 N and 8 N, respectively. It should be noted that this thickness offers acceptable initial adhesion, although its resistance seems to decrease with increasing loads, which could be the probable consequence of its small thickness compared with the larger thicknesses. The thickness at 450 nm (Figure 5B) shows a clear cohesive-type break and a perfectly straight line. There was a slight increase in the associated LC2 and LC3 critical loads, reaching around 7 N and 9 N, respectively, compared with the loads observed for the 250 nm thickness, while the initial critical load remained at around 4 N. This could suggest an improvement in mechanical strength and adhesion, due to a better distribution of stresses as a result of a thicker layer. For the 850 nm thick coating (Figure 5C), there was also a clear cohesive failure coupled with sequential displacement of the material from the indentor tip groove. Here, the critical loads are relatively similar to the 450 nm thickness, i.e., around 4 N, 7 N and 9 N. The most important fact observed here is the absence of adhesive-type fractures at the coating–substrate interface leading to coating delamination, irrespective of thickness, which generally reflects good adhesion of the coatings to their substrates.

Figure 6 shows scanning electron microscopy (SEM) images of the tracks after a scratch test on a 250 nm thick Ca-SZ coating compared to a track on the bare TA6V substrate. Figure 6A shows a straight track punctuated by a substantial accumulation of material carried by the indenter tip along the passage groove at the end of the track, suggesting a relatively low scratch resistance of the TA6V substrate (Figure 6B). In contrast, Figure 6C also shows a straight track and a clear cohesive break along the line. Figure 6D, at the end of the track, shows slight beads of material on either side of the track line, suggesting a protective effect of the coating on the underlying substrate when subjected to mechanical stress.

The EDS profiles of the scratch tracks obtained after a scratch test on Ca-SZ coatings of different thicknesses (250 nm, 450 nm and 850 nm) reveal notable variations in the elemental distribution as a function of coating thickness (Figure 7).

For the 250 nm coating, the titanium signal is particularly dominant along the entire scratch track, indicating a slight protective blanket and thus exposure of a significant portion of the underlying substrate to both the interaction depth and the excitation volume of the electron beam. The zirconium signal is irregular along the profile, suggesting partial abrasion of the coating. The detection of small amounts of oxygen correlates with the thinness of the coating. At 450 nm, the zirconium signal remains dominant along the profile, although there are signal drops at the edges of the track, reflecting more consistent coverage. Although titanium is still present, its signal is less pronounced than at 250 nm. The oxygen profile is also more constant, reflecting a more uniform distribution of the protective layer along the track. For the 850 nm coating, the zirconium signal is intense and continuous with no real interruptions, indicating greater coverage and resistance to wear. Oxygen is detected uniformly, suggesting that the substrate remains largely protected. This thickness appears to offer the best mechanical resistance, with a coating that remains covered even after mechanical stresses without significant exposure of the substrate. These observations confirm the correlation between increasing the thickness of the Ca-SZ coating and improving scratch resistance and the effectiveness of the underlying substrate protection.

## 4. Discussion

The results of this study highlight the influence of the thickness of Ca-SZ coatings on determining their intrinsic mechanical properties. SEM micrographs clearly show the absence of apparent defects, confirmed by EDS mapping, revealing a uniform distribution of constituent elements, irrespective of thickness, suggesting by correlation a controlled and reproducible PVD deposition process, crucial for guaranteeing mechanical and chemical stability for use in demanding clinical conditions. In addition, the roughness analysis shows a slight trend towards a non-significant increase in the Sa and Sq parameters in parallel with thickness, highlighting the coating’s ability to accommodate nanoscopic and microscopic irregularities while maintaining a uniform distribution. In this case, these observations are consistent with the principle of atom-by-atom deposition in PVD [19].

In addition, the UNHT measurements show a significant change in the apparent modulus of elasticity, ranging from 25.05 GPa to 43.35 GPa for 250 nm and 850 nm coatings, reflecting a direct causal link between the greater thicknesses and greater dissipation of elastic stresses under contact interaction in the thicker layers [20]. This behaviour could be explained by the reduced involvement of the underlying substrate at greater thicknesses. Conversely, it could be that the TA6V substrate, known for its Young’s modulus of 110 GPa, has a significant influence on the modulus of elasticity of the coating at 250 nm thickness, given its small thickness, a common observation in substrate–coating pairs where it is difficult to isolate the mechanical parameters of the coating correctly. From another point of view, it could simply be that the elastoplastic identities of the Ca-SZ coating are distinct and unique for each thickness. Nevertheless, this interrelationship between thickness and stress dissipation highlights the need for an optimum thickness that minimises substrate intervention under mechanical loading.

In any case, the mechanical adhesion tests show a slight gradual improvement in mechanical adhesion as the thickness of the Ca-SZ coating increases. In fact, a cohesive failure is reported over the entire track for the 250 nm thick coating, suggesting simple deformation and no delamination of the coating, reflecting the good mechanical strength of the coating–substrate pair. This observation is confirmed by the elemental EDS mapping of the track profile, revealing the effective presence of chemical elements making up the coating in the abrasion track, with the associated critical loads LC1, LC2 and LC3 being around 4 N, 6 N and 8 N, respectively. It should be noted here that the predominant titanium EDS signal throughout the tracks simply reflects the thinness of the coating, as this is much less than the interaction volume of the electron beam, which easily reaches the substrate at this depth [21]. At 450 nm, the fracture pattern is similar to the previous observation except for a slight increase in the associated LC2 and LC3 critical loads reflecting a reduction in the titanium signals observed in EDS mapping indicating better substrate protection compared with the 250 nm thickness. The almost complete coverage of zirconium along the tracks and similar critical loads observed in EDS mapping for the 850 nm coating correlates with the micrographic observation of the thickness track suggesting better substrate protection although the associated LC2 and LC3 critical loads are similar to the 450 nm thickness. It should be pointed out that although the increase in thickness predicts better resistance to wear and deformation, a structural limit seems to be reached beyond 450 nm, and is possibly linked to interactions between the substrate–coating pair or to intrinsic elastoplastic properties. From a mechanobiological point of view, the modulus of elasticity of the 250 nm and 450 nm thick Ca-SZ coatings approaches the values of the trabecular bone, with respective values of 14.8 GPa and a variation of 18.6 to 20.7 GPa for the cortical bone, rather than those of the underlying substrate [22,23]. This mechanical proximity could reduce stress shielding and improve osseointegration, following the principles of Frost’s manostat by promoting a homogeneous distribution of mechanical stresses, stimulating bone regeneration and minimising resorption [7,24,25]. Despite the encouraging results of this study, there are still areas to improve understanding. Further studies are needed to shed light on the singularity of the elastoplastic properties observed as a function of thickness variation, and further mechanical tests, such as pin–disk wear tests, micro-fretting and other coating performance tests, will therefore be essential to reinforce our conclusions and more accurately assess the durability and resistance of coatings in real-life conditions. A future study should investigate the impact of these coatings on cellular responses and the associated molecular mechanisms.

## 5. Conclusions

The aim of this study was to explore the interrelationship between thickness, stress dissipation and mechanical adhesion of the Ca-SZ coating. This study demonstrates that the thickness of the Ca-SZ coating plays a fundamental role in optimising its intrinsic mechanical properties, balancing wear resistance and adhesion to the substrate. The observed results contribute to a better understanding of the correlation mechanisms between mechanical resistance and film thickness, while raising the question of the ideal structural optimisation to maximise both biomechanical response and better adhesion to the substrate.

## Figures and Tables

**Figure 1 biomedicines-13-00037-f001:**
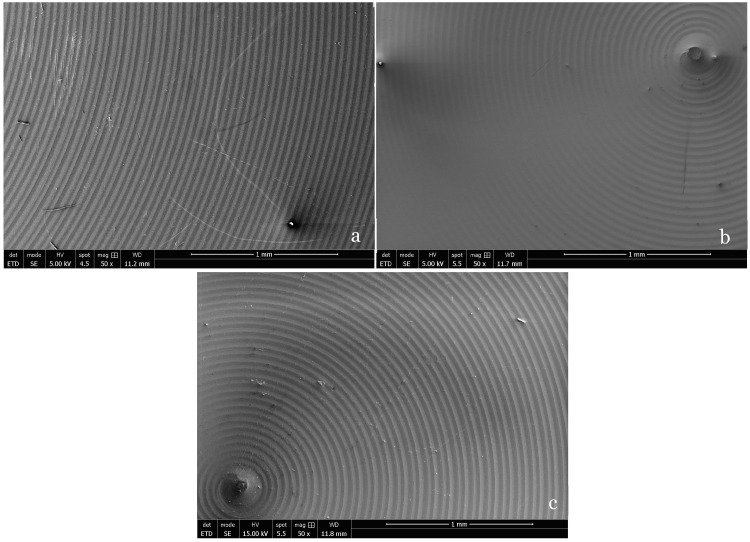
Three ×50 SEM micrographs of TA6V surfaces coated with 250 nm (**a**), 450 nm (**b**) and 850 nm (**c**) thick Ca-SZ (magnification ×50).

**Figure 2 biomedicines-13-00037-f002:**
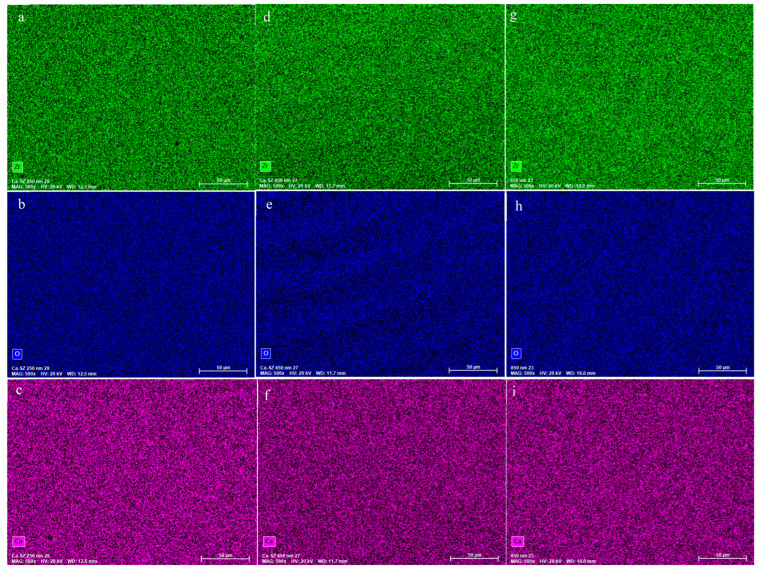
Elemental EDS maps of Ca-SZ coatings of thicknesses 250 (**a**–**c**) 450 (**d**–**f**) and 850 nm (**g**–**i**).

**Figure 3 biomedicines-13-00037-f003:**
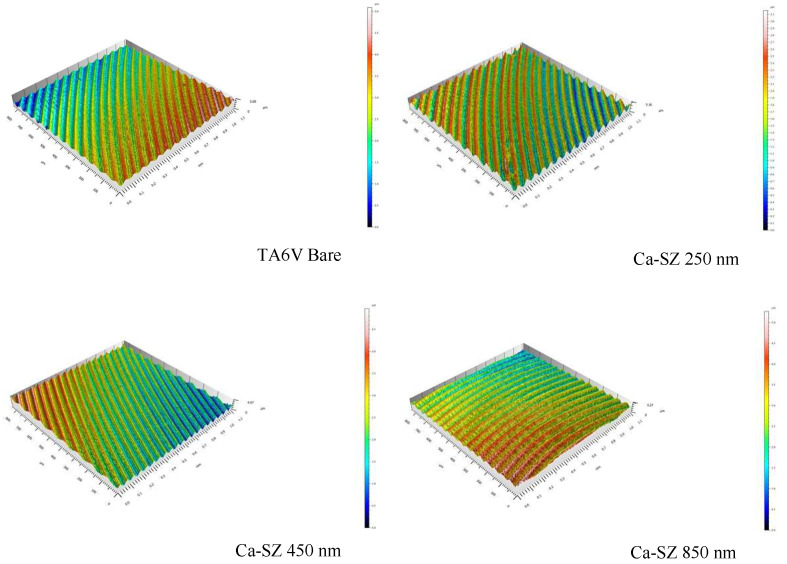
Three-dimensional surface profile of bare TA6V compared with TA6V surfaces coated with 250, 450 and 850 nm thick Ca-SZ.

**Figure 4 biomedicines-13-00037-f004:**
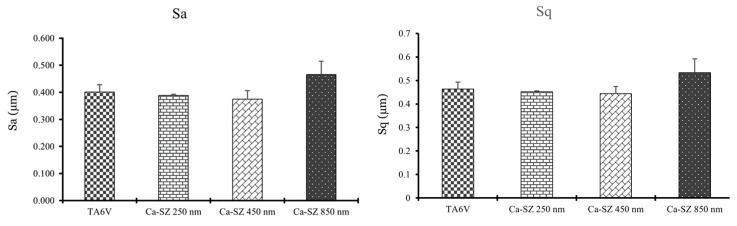
Evolution of topographic parameters Sa and Sq as a function of the thickness of the deposited Ca-SZ coating.

**Figure 5 biomedicines-13-00037-f005:**
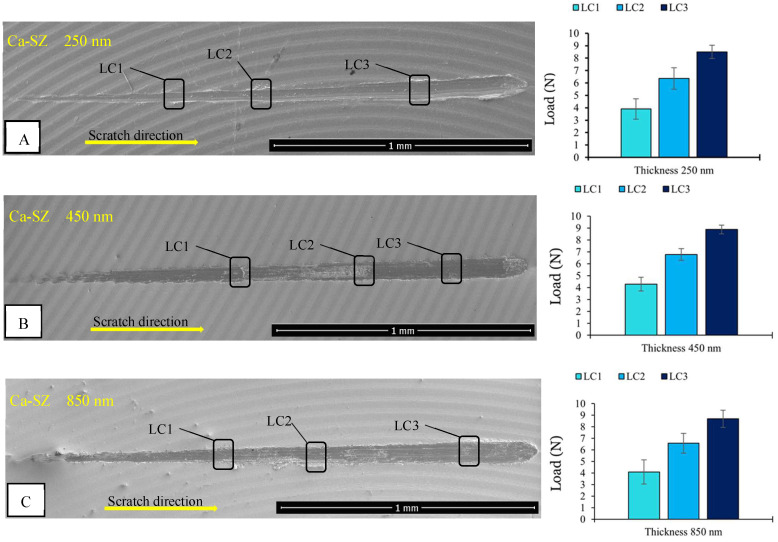
SEM micrograph after scratch test of the Ca-SZ coatings at thicknesses of 250 (**A**), 450 (**B**) and 850 nm (**C**) associated with the critical loads measured (LC1, LC2 and LC3) for each thickness.

**Figure 6 biomedicines-13-00037-f006:**
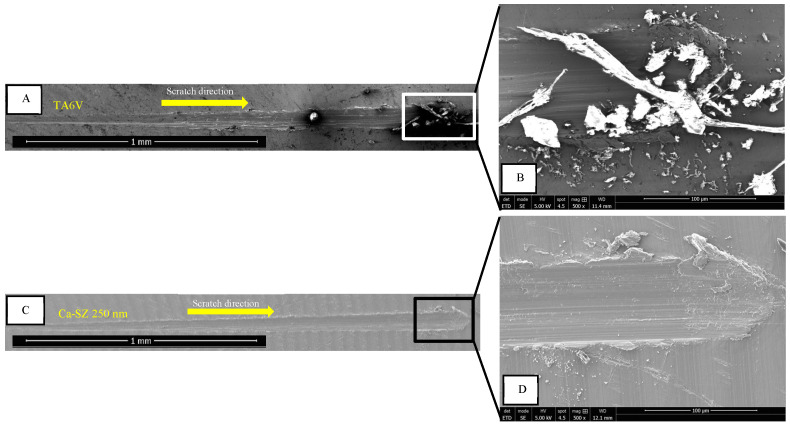
SEM micrograph after scratch test of a bare TA6V surface (**A**) compared with 250nm thick Ca-SZ coatings (**C**). (**B**,**D**) are end-of-track images at ×500 magnification.

**Figure 7 biomedicines-13-00037-f007:**
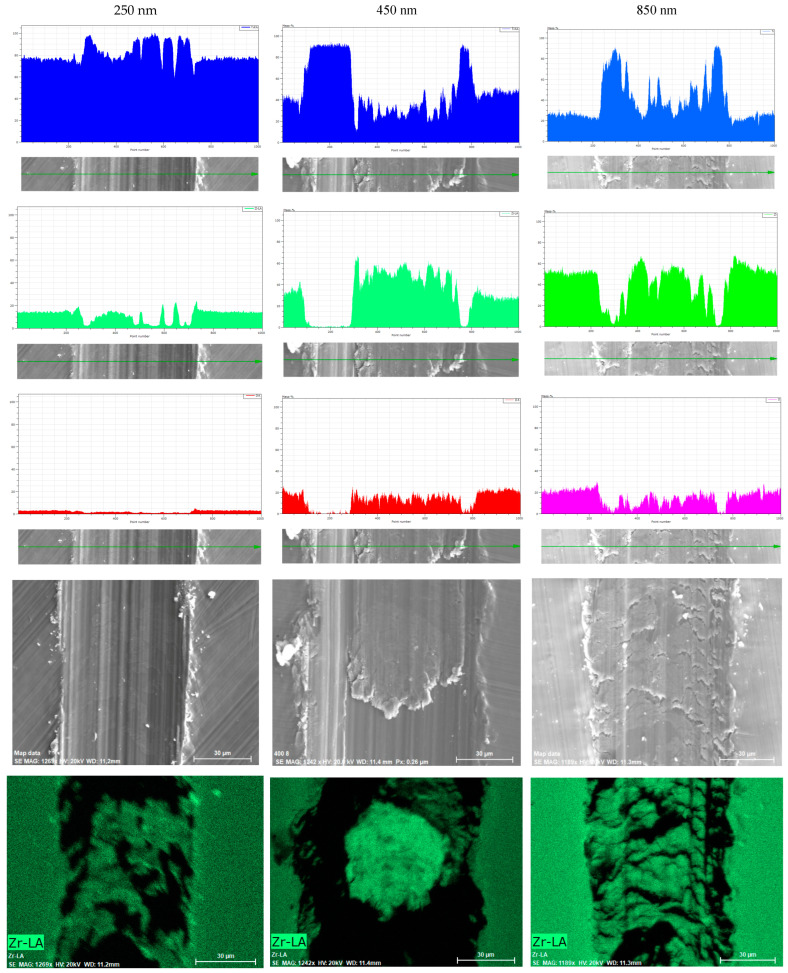
EDS profiles of scratch tracks after scratch testing on a 250, 450 and 850 nm thick Ca-SZ coating. The spectra show the elemental distributions of zirconium (green), titanium (blue) and oxygen (red for 250 and 450 nm, pink for 850 nm) along the track, illustrating the evolution of the coating’s coverage and resistance as a function of its thickness. The tendency of the track is indicated by the yellow arrow at the bottom of the column for each coating thickness.

**Table 1 biomedicines-13-00037-t001:** Evolution of Elastic Modulus with Coating Thickness of Ca-SZ.

Thickness	*k*1 (N/mm^2^)	*k*2 (N/mm^2^)	*k*3 (N/mm^2^)	Mean	*E* (GPa)
Ca-SZ 250 nm	37,360	37,395	35,402	36,719	25.05
Ca-SZ 450 nm	52,922	53,152	53,699	53,257.6	36.3
Ca-SZ 850 nm	62,922	65,952	64,777	64,550.3	43.35

## Data Availability

The original contributions presented in this study are included in the article. Further inquiries can be directed to the corresponding author.

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
