# Peer review of "Multi-Scale Characterisation and Mechanical Adhesion in PVD-Deposited Ca-SZ Coating for Implantable Medical Devices"

_biomedicines, 2024, doi:10.3390/biomedicines13010037_

Round 1
Reviewer 1 Report
Comments and Suggestions for Authors
General remarks
1. Trademark symbols should be removed throughout the entire manscript in order to avoid a mercantile appearance.
2. Scale bars, particularly in SEM and EDS images need to be magnified, as they are barely recognizable in most cases.
M&M section
1. Briefly decribe in section 2.1. the applied method, rather than only referring to an external source.
2. In section 2.2., first line: The expression "for some" appears strange and unclear, please rephrase.
3. In section 2.3, parameters Sa and Sq, which are later evaluated, should be briefly introduced and defined.
Results section
1. In section 3.3, major parts of lines 132-139, which are describing methodology, should be moved to M&M section 2.4.´, where also parameters k1-3 should be briefly introduced and decribed.
2. In section 3.4, the parameters LC1-3 are introduced without any prior mentioning in the respective M&M section 2.5. This need to be corrected.
3. Axes labels in Fig. 7 are unrecognizable, see also general remarks, and need to be magnified. In the SEM parts of the figures, scale bars should be introduced in order to indicated the length of the yellow arrows. Also, 2D EDS maps, similar to Fig. 2, and maybe in comparable maginifaction, should be added to Fig. 7, as they would strengthen the results and be helpful to the reader.
Discussion section
1. Parameters Sa and Sq are being referred to, but they are not defined in the M&M section; see also comment above.
2. Discussion of Ca-SZ coating thickness dependent elastic modulus, lines 221-233, so far only relates to a specific value for TA6V in line 227. It is highly advisable to also include a specific value for pure/bulk Ca-SZ. The reviewer is aware that, depending on crystalline architecture, as alluded to in lines 230/231, Ca-SZ may span a broad range of elastic moduli. Hence, this elastic modulus range of Ca-SZ of SZ should be discussed and, in case own data is absent, literature data should be included into the discussion.
Reviewer 2 Report
Comments and Suggestions for Authors
Thank you for the opportunity to review this manuscript. I appreciate the effort that the authors have put into their work. While your manuscript presents interesting findings regarding the mechanical properties of Ca-SZ coatings, several major concerns need to be addressed before the manuscript can be considered for publication.
Major issues:
The primary concerns relate to unsupported claims and inappropriate extrapolations of mechanical data to biological outcomes.
1- Current claim (Line 67-68): "...the presence of zirconia on the surface also acts as an effective shield to limit bacterial biofilm colonisation and proliferation, thereby eliminating the risk of peri-implantitis". However, no experimental data is provided to support this claim
2- Current claim (Lines 64-66): "...calcium doping of zirconia...aims to functionalise surfaces by chemotaxis activation of osteocytes responsible for bone remodelling". No experimental evidence provided for chemotaxis or osteocyte activation
3- Current issue: Claims of "better" performance at 850nm despite similar critical loads to 450nm
Please:
* Provide clearer criteria for "better" performance
* Include statistical analysis of differences between thicknesses
* Revise conclusions to accurately reflect the measured parameters
4- Current claim: Conclusions about biomechanical compatibility without direct testing
5- Current issue: Claims about improved osseointegration without supporting evidence
Minor issues:
1. Add a clear "Limitations" section addressing:
2. Please revise the following sections:
A. Abstract: Remove speculative claims about biological outcomes
B. Introduction: Clarify the focus on mechanical properties
C. Discussion: Separate measured outcomes from hypothetical benefits
D. Conclusion: Limit to actual findings rather than potential applications
Round 2
Reviewer 2 Report
Comments and Suggestions for Authors
Please update references 10 and 11 in the document. I would like to have more recent sources that are relevant to the topic discussed. If possible, provide a brief summary of the new references.
Comments on the Quality of English Language
Is it ok
Author Response
The mentioned references have been updated, and more recent references have been added in accordance with your recommendations.